# Federated End-to-End Unrolled Models for Magnetic Resonance Image Reconstruction

**DOI:** 10.3390/bioengineering10030364

**Published:** 2023-03-16

**Authors:** Brett R. Levac, Marius Arvinte, Jonathan I. Tamir

**Affiliations:** 1Chandra Family Department of Electrical and Computer Engineering, The University of Texas at Austin, Austin, TX 78712, USA; 2Intel Corporation, Hillsboro, OR 97124, USA; 3Dell Medical School Department of Diagnostic Medicine, The University of Texas at Austin, Austin, TX 78712, USA; 4The Oden Institute for Computational Engineering and Sciences, The University of Texas at Austin, Austin, TX 78712, USA

**Keywords:** MRI reconstruction, federated learning, unrolled optimization, accelerated MRI

## Abstract

Image reconstruction is the process of recovering an image from raw, under-sampled signal measurements, and is a critical step in diagnostic medical imaging, such as magnetic resonance imaging (MRI). Recently, data-driven methods have led to improved image quality in MRI reconstruction using a limited number of measurements, but these methods typically rely on the existence of a large, centralized database of fully sampled scans for training. In this work, we investigate federated learning for MRI reconstruction using end-to-end unrolled deep learning models as a means of training global models across multiple clients (data sites), while keeping individual scans local. We empirically identify a low-data regime across a large number of heterogeneous scans, where a small number of training samples per client are available and non-collaborative models lead to performance drops. In this regime, we investigate the performance of adaptive federated optimization algorithms as a function of client data distribution and communication budget. Experimental results show that adaptive optimization algorithms are well suited for the federated learning of unrolled models, even in a limited-data regime (50 slices per data site), and that client-sided personalization can improve reconstruction quality for clients that did not participate in training.

## 1. Introduction

Magnetic resonance imaging (MRI) is a medical imaging modality that has seen widespread clinical adoption due to its superior soft tissue contrast and use of non-ionizing radiation. In particular, multi-coil (parallel) MRI acquisition combined with sub-sampling represents a means of reducing scan time and increasing scanner throughput [1,2,3]. Given that sampling below the Nyquist rate gives rise to an ill-posed inverse problem, and hence can produce artifacts in reconstructed images, a wide area of research has been dedicated to image reconstruction algorithms for accelerated MRI. Recently, deep learning has achieved substantial improvements in reconstruction quality compared to classical methods [4,5,6], and is rapidly heading toward clinical validation [7].

One particular challenge that data-driven MRI reconstruction methods face is the availability of large datasets containing raw signal measurements (called k-space) acquired at a specific data site (client) in a multi-clinic setting. As independent institutions are often equipped with MRI scanners supplied by different vendors, together with different clinical needs in terms of patient population and institutional scanning protocols, this leads to a data heterogeneity problem, where a particular site may have both disparate and insufficient data to train a high-fidelity model. This has motivated researchers to investigate the transfer capabilities of deep learning for MRI reconstruction [8] and led to the use of federated learning [9]—a collaborative learning technique which leverages decentralized training—for medical image reconstruction in a cross-silo setting. Given data scarcity issues, together with regulatory and privacy constraints that prevent raw data from being shared between institutions [10], federated learning is a promising tool for achieving robust generalization. Finally, models learned using federated learning have the potential to be further personalized [11,12], where models are adapted at the client side after learning has concluded in order to improve performance on the local sample distributions. One such application is for sites that may only be capable of acquiring and processing a limited number of fully sampled scans (e.g., due to specialized system hardware or limited resources) to also benefit from the learned model without having participated during training in the first place.

End-to-end unrolled optimization methods [13] are considered state of the art for under-sampled MRI reconstruction [8], and are the focus of this work. We investigate established adaptive federated learning algorithms applied to 2D (per slice) MRI reconstruction with end-to-end unrolled optimization. In the main federated learning stage, an adaptive optimization algorithm is used to learn an MRI reconstruction model without sharing data or features between clients. Subsequently, in the personalization stage, a client uses the small number of available fully sampled MRI scans (as few as 50 2D slices) to fine tune the global model on the target distribution using cross-validation early stopping. We perform experiments using subsets from the fastMRI [14] knee and brain datasets, as well as publicly available axial knee and abdominal datasets [15], and demonstrate that adaptive federated learning algorithms can efficiently learn reconstruction models across a wide range of anatomies, contrasts, and communication rates, and are also amenable to personalization via simple fine tuning.

### 1.1. Background

#### 1.1.1. Federated Learning for MRI Reconstruction

Federated learning [16] has recently gained substantial research interest in the broader machine learning field [17] due to its ability to learn models in a distributed and privacy-oriented manner. A baseline federated optimization algorithm is given by federated averaging (FedAvg) [16], where individual clients periodically upload their model weights to a central server, which performs simple averaging of the client-sided model weights. Powerful, FedAvg performance is known to degrade when data across clients are heterogeneous. The work in [18] introduces the Scaffold algorithm as a solution to learning shared representations for heterogeneous data by including an adaptive momentum term at each communication round. Similarly, the work in [19] proposes a family of adaptive optimization algorithms (termed FedAdam, FedAdaGrad, and FedYogi after their classical counterparts) for federated learning. In this work, we investigate the performance of these adaptive algorithms when applied to unrolled MRI reconstruction with deep networks.

Federated learning has recently been applied to medical imaging in the context of a synchronous, cross-silo setting, and is foreseen to have an impact on the future of digital health [10]. Important challenges when dealing with medical image data are given by data heterogeneity [20,21], as well as the sensitive nature of raw patient data [22]. When applied to MRI reconstruction, the FL-MRCM algorithm [9] proposes an adversarial feature learning approach for handling the heterogeneous nature of the data. In this method, one of the clients is designated as an explicit target, and the feature representations of all other clients are optimized to be indistinguishable from that of the target. The recent work in [23] used generative adversarial networks (GANs) to build on the FedGAN approach [24] and proposed learning a deep generative prior using federated learning that is further personalized for each individual scan during test time. However, it is an open question if adversarial learning approaches are suitable for scenarios with very limited training data at client sites. The work in [25] proposed a model splitting approach inspired by the layer personalization method in [26] but this is only suitable for non-unrolled architectures, such as U-Net [27]. The work in [28] discussed strategies for splitting unrolled reconstruction models in centralized multi-task settings, but leaves as future work to investigate these in conjunction with federated learning and heterogeneous clients.

Client-sided personalization using fine tuning was previously introduced in [29], where the authors investigated the efficiency of this approach on language tasks. The work in [30] introduced a framework based on knowledge distillation to personalize the models in the final communication round in computation-bound settings. More recently, the work in [31] provided theoretical justification for fine tuning in the context of linear models and the FedAvg algorithm. Inspired by this line of work, we also investigate the performance of fine tuning as a simple and efficient method of personalizing MRI reconstruction models.

#### 1.1.2. Unrolled Optimization for MRI Reconstruction

Model-based unrolled optimization architectures represent a large area of research in ill-posed inverse problems [13,32], including medical imaging. The work in [4] introduced the variational network (VarNet) approach, where differentiable optimization steps are interleaved with a forward pass of a deep neural network, and the entire architecture is trained end-to-end using a supervised reconstruction objective. A similar method was given by the model-based deep learning (MoDL) approach [33], with differences in the type of unrolled optimization used. Extensions of these methods have been proposed to include automatic sensitivity map estimation [34,35,36], dual domain (image and k-space) formulations [37,38], and self-supervised settings [39].

While these works have greatly advanced MRI reconstruction benchmarks [8], there are still open questions regarding how they scale in a federated learning regime with few local samples, as well as the best approach for handling domain shift [21]. Our work aims to address the current research gap that exists at the intersection of federated learning and unrolled end-to-end optimization for MRI reconstruction. We focus on the baseline MoDL approach introduced in [33], and investigate its performance in a federated setting, under different optimization algorithms and communication rates. We consider a multi-coil learning setup using realistic k-space measurements. This is in contrast to other works, where some of the measurements are simulated from coil-combined images. While the latter allows for simple implementations of deep learning methods, the results may not directly translate to a clinical settings [40]. We investigate reconstruction performance in federated learning regimes with a small number of samples (slices) available for local training or personalization (as low as 50). Previous work in federated MRI reconstruction only considered up to four clients during learning [9,23], each having access to at least hundreds of training samples. While this is a valid setup for cross-silo federated learning, acquiring large amounts of data in under-resourced clinics may be untenable. In addition, previous work has shown that this data regime may already be sufficient to train unrolled models for MRI [41], which we confirm in our experiments.

### 1.2. Contributions

A summary of our contributions in this work is the following:We perform extensive experimental evaluations to determine the low data regime of federated learning end-to-end unrolled MRI reconstruction. This is the regime where each client has an insufficient number of samples to accommodate local (non-collaborative) learning, enabling federated learning to benefit every participant. We find that, across a wide range of anatomies and contrasts, this regime consistently occurs when fewer than 50 slices (across five patients) are available at each site.We evaluate non-adaptive and adaptive federated learning algorithms (FedAvg, FedAdam, FedAdagrad, FedYogi, and Scaffold) applied to unrolled MRI reconstruction in the low data regime. We investigate both independent and identically distributed (i.i.d.) and non-i.i.d. client settings, as well as their performance as a function of the frequency of communication in a setting with fixed total computational power. Our findings indicate that federated unrolled optimization for MRI reconstruction is feasible with as few as four infrequent communication rounds.We evaluate a client-sided model personalization via fine tuning after federated learning, using a small amount of fully sampled MRI scans. We find that this can learn an improved reconstruction network without heavily overfitting in the low-data regime, even for clients that did not originally participate in training. Fine tuning constitutes an easy, reproducible benchmark for future federated learning researchers to build and improve on.

The code for the experiments in this paper is publicly available (https://github.com/utcsilab/Unrolled_FedLrn, accessed on 6 February 2023).

## 2. Theory

### 2.1. System Model

Two-dimensional MRI operates by measuring the k-space (spatial frequency) domain of a complex-valued image $x\in C^{N}$, where we use lowercase letters to indicate the vectorized version of a multidimensional data array, and *N* denotes the length of the vectorized data. Letting $F\in C^{N\times N}$ be the two-dimensional Fourier operator matrix, and $P\in {\left\{0,1\right\}}^{M\times N}$ be a binary selection (diagonal) matrix, the noisy, under-sampled vector of measurements $y\in C^{M}$ for the case of single-coil MRI with Cartesian sampling is given by
(1)y=PFx+n,
where *n* is (without loss of generality) zero-mean complex-valued white Gaussian noise.

Multi-coil MRI measures the same image *x* with an array of radio-frequency receive coils, each with a spatially varying sensitivity profile, producing a set of parallel measurements $y_{i}$, where $i\in \{1,\ldots ,N_{c}\}$, and $N_{c}$ represents the number of receive coils. Each coil is characterized by the coil sensitivity map diagonal matrix $S_{i}\in C^{N\times N}$, and the forward model for a single coil is given by [2]
(2)$$y_{i}=PFS_{i}x+n_{i}.$$

We define R=M/N as the acceleration factor. Importantly, because the receive coil sensitivity profiles generally vary smoothly, measurements acquired from different coils are correlated, making the inverse problem of recovering *x* from multi-coil measurements potentially ill-posed, even at relatively low *R*. For the remainder of this work, we abuse notation and let *y* and *n* denote the concatenated vectors of multi-coil measurements and additive noise, respectively, *S* the concatenated matrix of sensitivity maps, and A=PFS the summarized forward operator that encompasses all three operations in sequence. We obtain that the multi-coil forward MRI model is given by
(3)y=Ax+n.

The goal of MRI reconstruction is to recover the image *x* from the under-sampled measurements *y*, and can be formulated as the solution to the following optimization problem:(4)$$\underset{x}{\mathrm{argmin}}\frac{1}{2}{∥y-Ax∥}_{2}^{2}+\lambda R\left(x\right),$$
where R and λ are a suitably chosen prior and regularization coefficient, respectively. Designing or learning the correct prior is a critical aspect in recovering signals from under-sampled measurements. For example, compressed sensing based recovery [42] utilizes $R\left(x\right)={∥Wx∥}_{1}$, where *W* is a wavelet operator, while deep learning methods use a learned prior either explicitly or implicitly represented by a neural network D(·;Θ) with learnable weights Θ [4,33,34,35,43]. We note that this regularized linear inverse problem formulation is also applicable to other imaging modalities, such as computed tomography, making signal processing methods that work for one medical imaging modality potentially applicable to others.

### 2.2. MRI Reconstruction with End-to-End Unrolled Models

End-to-end unrolled models are a family of powerful, data-driven recovery algorithms that have been successfully applied to medical imaging, as well as to other areas of computational science [6]. In the centralized setting, a large corpus of training samples are available at a central location, and an unrolled model is trained end to end. When different sites have different data distributions, then the centralized algorithm is assumed to have access to data from all sites. In this work, we consider the setting of MoDL [33], where image reconstruction is formulated as the solution to the optimization problem:(5)$$\arg\min_{x}{∥y-Ax∥}_{2}^{2}+\lambda {∥D\left(x;\Theta \right)-x∥}_{2}^{2},$$
where $D\left(\cdot ;\Theta \right)$ is a deep neural network with weights Θ. Using an alternating minimization approach leads to the following iterative algorithm [33] for approximately solving Equation (Equation 5):(6)$$z_{n+1}=\arg\min_{z}{∥y-Az∥}_{2}^{2}+\lambda {∥x_{n}-z∥}_{2}^{2},$$
(7)$$x_{n+1}=D\left(z_{n+1};\Theta \right).$$

The solution to (Equation 6) can be expressed in closed-form as
(8)$$z_{n+1}={\left(A^{H}A+\lambda I\right)}^{-1}\left(A^{H}y+\lambda x_{n}\right),$$
where $A^{H}$ is the Hermitian transpose (adjoint) of *A*. In practice, due to the large variable sizes, the solution to (Equation 8) is approximated using a finite number of iterations of the conjugate gradient (CG) algorithm [44]. The overall image reconstruction process alternates between a number of iterations of CG, followed by passing the resulting image through the network D, and repeated for a number of $N_{u}$ unrolls, yielding the output image $x_{N_{u}}$. The weights Θ are trained using a supervised image reconstruction loss. In this work, we use the structural similarity index (SSIM) [45] for training as
(9)$$L_{\mathrm{train}}\left(\Theta \right)=-E_{x,y}\left[\mathrm{SSIM}\left(\mathrm{RSS}\left(x_{N_{u}}\right),\mathrm{RSS}\left(x\right)\right)\right],$$
where the RSS function consists in applying the sensitivity map operator *S* and taking root-sum-of-squares (RSS) along the coil axis to obtain a magnitude-only image, and $\mathrm{RSS}\left(x\right)$ is the RSS obtained after an inverse Fourier transform of the fully sampled measurements. The expectation is taken over a dataset of ground truth images *x*, and retrospectively under-sampled measurements *y*.

### 2.3. Federated Unrolled Optimization

In the federated learning setting, clients collaborate to train a global MRI reconstruction model without sharing raw data (such as k-space and RSS images), or features (such as derived representations of the data). Instead, a number of *K* clients upload model weights $\Theta_{k}^{i}$ after training local models for a given number of local optimization steps in the *i*-th *round*, which are aggregated by the central server to yield $\Theta_{G}^{i}$, and broadcast back to clients. For the remainder of this work, we make the following two assumptions:**Synchronicity**: Clients optimize their local models for the same number of steps, and upload their weights to the server synchronously.**Full participation**: After every round, all clients upload their weights to the server.

A schematic of the federated learning process is shown in Figure 1. The two assumptions above are common in the cross-silo federated learning setting [17], which may match clinical settings well. The server can use any meaningful update rule to aggregate the weights. A baseline method is given by simple weight averaging in the form of the FedAvg algorithm [16]:(10)$$\Theta_{G}^{i}=\frac{1}{K}\sum_{i=1}^{K}\Theta_{k}^{i-1}.$$

Adaptive federated optimization algorithms seek to prevent client drift and improve convergence rates [17,18,19], which is relevant in settings with non-i.i.d. clients. In general, these methods introduce auxiliary variables which are updated along with the network weights. At a high level, the server performs weight aggregation using a functional *f* and auxiliary variables ϕ in the form
(11)$$\Theta_{G}^{i}=f\left(\Theta_{1}^{i-1},\ldots ,\Theta_{K}^{i-1};\phi \right).$$

When applied to unrolled optimization, we assume that each client trains, uploads, and downloads the regularization model $D(\cdot ;\Theta_{k}^{i})$. After the *R*-th and final round is completed, the server broadcasts the final model $D_{G}\left(\cdot ;\Theta_{G}^{R}\right)$ to all clients, who then apply $N_{u}$ iterations of (Equation 6) and (Equation 7) during inference.

## 3. Methods

### 3.1. Datasets and Model Architecture

We use three publicly available datasets in the experiments: fastMRI [14], abdominal scans from [15], which we refer to as the Stanford dataset, and axial knee scans from [15], which we refer to as the NYU axial knee dataset. We use fastMRI for training due to its large number of available contrasts and field strengths, which allows us to simulate a federated scenario with heterogeneous client data distributions. In particular, we simulate a scenario with ten clients, each with different combinations of anatomy (knee and brain), contrast (PD, PDFS, T1, T2, and FLAIR), and field strength (1.5T and 3T). We use the Stanford and NYU axial knee datasets, in addition to two other anatomy and contrast combinations from fastMRI, as means of investigating reconstruction performance of clients that did not participate in the federated learning.

For all experiments, we assume an acceleration factor R=4 with a random under-sampling mask along the phase encode (PE) direction and a fully sampled central region, which was taken as 8% of the total PE lines present. As the readout direction of the Stanford abdomen is horizontal, we first rotate k-space prior to reconstruction, and rotate back after. All data are normalized by the maximum value of the RSS image computed using the fully sampled central lines of k-space. We use the ESPIRiT algorithm [46] in the BART toolbox [47] to estimate the coil sensitivity maps. We use an MoDL architecture consisting of $N_{u}=6$ unrolls. Data consistency blocks are implemented with $N_{CG}=6$ steps. The regularization network is a U-Net architecture which takes two input channels (real, imaginary), and consists of 481,092 trainable weights. Local clients were updated using the Adam optimizer [48] with a batch size of one.

### 3.2. Low Data Regime for Federated Learning

As federated training of MRI reconstruction models aim to solve the issue of limited amounts of fully sampled data for some institutions, we first aim to determine the local sample count, which characterizes this notion of “limited data” for our setting. Specifically, we are interested in the number of local samples that causes a considerable drop in local performance but is still reasonably large to enable federated learning gains. To quantify this drop, we train local models using a variable amount of local training samples, without any federated learning involved, and empirically determine this dropping point. Specifically, we train centralized MoDL networks on each data distribution listed in Table 1 (separate models for each) using a varying number of fully sampled training scans.

### 3.3. i.i.d. vs. Non-i.i.d. Client Distributions

Different client data distributions are a key issue that impacts the convergence of federated learning optimization, and are also highly common to MRI. In this section, we describe the methodology used to investigate the impact of heterogeneous, differently distributed clients on federated end-to-end unrolled approaches, and whether adaptive algorithms can mitigate these shifts. We first investigate the performance when all ten clients are composed of non-overlapping PDFS 1.5T scans (site 2) from the fastMRI dataset. In the case of non-i.i.d. client distributions, we sample the local data of each client from one of ten different distributions (sites), as summarized in Table 1. This leads to no two clients sharing the same anatomy, contrast, or field strength combination, and is a realistic cross-silo federated learning setting. In both cases, we perform synchronous, full-participation federated learning for 240 aggregation rounds, where each round consists of two local epochs of training (100 local optimization steps).

We evaluate the following four adaptive federated optimization algorithms: Scaffold, FedAdam, FedYogi, and FedAdaGrad. We also evaluate the FL-MRCM approach in [9], where we include the two additional Sites 4 and 12 in the federation for the non-i.i.d. case, and designate them as targets for the adversarial objective.

*Scaffold* [18] is summarized in Algorithm 1 and uses auxiliary variables $c_{g}$ and $c_{k}$ that represent global and local momentum terms, respectively. During client optimization, their difference can be seen as an estimate of the client drift, and they are incorporated in the local updates (line 7 in Algorithm 1) to counteract this.*FedAdam*, *FedYogi*, and *FedAdaGrad* [19] are the federated versions of their namesake centralized algorithms.The *FL-MRCM* approach in [9] proposes a solution to the problem of non-i.i.d. clients in federated learning by using an adversarial loss on the feature space of each client’s local reconstruction network. A key idea of the approach is to designate a specific client as a target, and to share its feature representations of the data with all other clients, where an adversarial feature objective between the local and the target client’s is used to make the feature representations as similar as possible.

**Algorithm 1** *Scaffold* [18]**Global inputs:** Initial $\Theta_{G}^{0}$, $c_{g}^{0}$, and global step size $\eta_{g}$.**Client inputs:** Local client datasets $D=\{D^{1},\ldots ,D^{K}\}$, initial $c_{k}^{0}$, and local step size $\eta_{l}$.**Output:** Global model weights $\Theta_{G}^{R}$.
  1:**for** round i=1,…,R **do**  2:   **download** $\Theta_{G}^{i}$ and $c_{g}^{i}$ for all *K* clients  3:   **for** client *k* in *K* **do**  4:         **for** optimization step s=1,…,S **do**  5:          Compute gradient $g_{k}\left(\Theta_{k}^{i}\left(x_{s}\right)\right)$  6:          $\Theta_{k}^{i}\overset{}{\leftarrow }\Theta_{k}^{i}-\eta_{l}\left(g_{k}\left(\Theta_{k}^{i}\left(x_{s}\right)\right)-c_{k}^{i}+c_{g}^{i}\right)))$  7:         $c_{k}^{+}\overset{}{\leftarrow }$ (*i*) $g_{k}\left(\Theta_{k}^{i}\right)$, or (*ii*) $c_{k}^{i}-c_{g}^{i}+\frac{1}{S\eta_{l}}\left(\Theta_{G}^{i}-\Theta_{k}^{i}\right)$  8:         **upload** $\left(\Delta \Theta_{k}^{i},\Delta c_{k}^{i},c_{k}^{i}\right)\overset{}{\leftarrow }\left(\Theta_{k}^{i}-\Theta_{G}^{i},c_{k}^{+}-c_{k}^{i},c_{k}^{+}\right)$  9:   $\left(\Delta \Theta_{G}^{i},\Delta c^{i}\right)\overset{}{\leftarrow }\frac{1}{K}\sum \left(\Delta \Theta_{k}^{i},\Delta c_{k}^{i}\right)$10:   $\Theta_{G}^{i+1}\overset{}{\leftarrow }\Theta_{G}^{i}+\eta_{g}\Delta \Theta_{G}^{i}$ and $c_{g}^{i+1}\overset{}{\leftarrow }c_{g}^{i}+\Delta c^{i}$


### 3.4. Personalized Unrolled Optimization

After a number of *R* communication rounds are completed, the server stops requesting weight uploads and communicates $\Theta_{G}^{R}$ to all clients. This is a global model, trained to reconstruct multiple, potentially heterogeneous images resulting from differences in scan protocol, imaging anatomy, or system hardware. Therefore, there are two prevailing issues with directly using the global model that we study in this work:Reconstruction performance may not be satisfactory for certain clients that participated in the federated learning process because characteristics of their image distributions (such as anatomy or contrast) are under-represented, hence client-sided personalization is an efficient tool for boosting performance.New clients that do not have access to large training datasets may not benefit by from only using the pre-trained models provided by federated learning, if their data types are not well represented in the original federation of clients. This makes client-sided personalization a necessary component for acceptable quality reconstructions.

To address the above issues, we test the impact of client-sided personalization through fine tuning, which has recently been shown to be a competitive approach for improving performance on a local dataset [31]. Formally, fine tuning is done with the same SSIM loss as in (Equation 9) and is summarized in the right side of Figure 1 for a new client. There are two important hyper-parameters that control personalization through fine-tuning: the learning rate $r_{\mathrm{fine}}$, and the number of fine-tuning epochs $N_{\mathrm{fine}}$. To select these hyper-parameters, we propose that each client perform a five-fold cross-validation approach using only its own local training data. We then use all available local client data and the selected hyper-parameter values to fine tune the federated model to the local client distribution.

We also investigate the impact of the communication budget on the personalization results after training. We focus on the non-i.i.d. case with Scaffold. In each scenario, the same total number of local update steps is allocated (24,000 local steps), except now a varying number of federated communication rounds (240, 120, 80, 60, 40, 24, or 4) is allowed. After learning the federated models, we test reconstruction performance both on clients that participated in the federated training and new, unseen clients.

### 3.5. Quantitative Metrics

We evaluate quantitative reconstruction performance using SSIM as defined in (Equation 9) (higher is better) and normalized root mean squared error (NRMSE) between the RSS reconstruction and the fully sampled RSS (lower is better), where NRMSE between a reference *x* and reconstruction $\hat{x}$ is defined as
(12)$$\begin{array}{cc} \mathrm{NRMSE}\left(x,\hat{x}\right) & =\frac{{∥x-\hat{x}∥}_{2}}{{∥x∥}_{2}}. \end{array}$$
Tables reporting SSIM and NRMSE represent the average value of the metric on the validation set.

## 4. Results

### 4.1. Finding the Low Data Regime

The impact of number of training samples on reconstruction performance is shown in Figure 2. When at least 250 local training samples are available to the unrolled model at training time, the performance saturates, and additional training samples have marginal impact on the validation performance. In contrast, training using 50 local scans leads to a decrease in performance across different sites. Using this information, we chose 50 local samples for each client in the remainder of experiments. A similar plot is provided for a U-Net reconstruction architecture in Figure A1 in Appendix A. We observe that SSIM and NRMSE should not be directly compared across different sites due to different points of performance saturation, even in the high-data regime. This is well documented, as the reconstruction quality is heavily influenced by anatomy, contrast, and field strength [21].

### 4.2. i.i.d. vs. Non-i.i.d. Clients

Based on the findings in Section 4.1, we selected 10 slices each from five different subjects, for a total of 50 local slices per client from the fastMRI dataset—this holds both for the i.i.d. and non-i.i.d. settings. The result of i.i.d. training is summarized in Table 2, which shows the average performance across all 10 sites that participated in training. For brevity, we report the average as we found that i.i.d. performance generally matched centralized training. All federated methods perform about the same or slightly better than centralized training in SSIM. The same is true with NRMSE. Centralized training and the federated unrolled optimization methods outperformed FL-MRCM. Figure 3 shows an example reconstruction for the i.i.d. case, comparing the ground-truth (GT) to FedAvg, Scaffold, FL-MRCM, and centralized training. Scaffold performed slightly better than FedAvg for this particular slice, and on-par with centralized, while FL-MRCM is poor, consistent with the results in Table 2.

The results of non-i.i.d. experiments are summarized in Table 3 and Table A1 in Appendix A for all fastMRI sites that participated in training. In this case, centralized and FedAvg perform about the same, while the adaptive federated algorithms typically perform slightly better. FL-MRCM is not competitive in this regime. Figure 4 shows example reconstructions of slices from two different sites. In this case, there is a clear qualitative and quantitative improvement between Scaffold and FedAvg, where the latter performs on par with centralized training.

### 4.3. Personalization and Impact of Communication Budget

As mentioned in Section 3.4, after training, we personalized the model at each new site through fine tuning. We tuned two client-specific hyper-parameters ( $r_{\mathrm{fine}}$ and $N_{\mathrm{fine}}$) at one seen site (site 2) and two unseen sites (Stanford and NYU axial knee) using a five-fold cross-validation scheme on the available 50 local slices. The resulting hyper-parameters are displayed in Table 4. We picked site 2 (fastMRI, fat suppressed knee, 3T) from the sites present during training because it comes from the anatomy less represented among all clients, leaving room for more personalization gains similar to the unseen sites during training. Exemplar reconstructions are shown for one out-of-distribution client (Stanford abdomen) at 240 communication rounds and one in-distribution client (site 2) at four communication rounds in Figure 5 and Figure 6, respectively. In both cases, there is a substantial drop in performance for Scaffold, which is recovered after fine tuning.

Quantitative reconstruction results as a function of communication rounds are shown in Figure 7 for three different clients using: (i) models only trained using Scaffold federated learning (Scaffold), and (ii) models trained using Scaffold federated learning, followed by personalization on local data (Scaffold + personalization). The exemplar clients that were chosen are site 2, NYU axial knee, and Stanford abdomen. We chose site 2 (fastMRI knee, fat-suppressed) to contrast the performance, and optimal hyper-parameters, of a site seen during training and sites unseen during training (NYU axial knee, Stanford abdomen). Results for additional sites are shown in Figure A2.

We note that model personalization via fine-tuning displays major benefits primarily for under-represented clients in federated training (i.e., knee scans), as well as those unseen (that did not participate) during federated training. For clients who are seen during training, we note that as the communication budget decreases (less frequent weight sharing), test performance decreases noticeably in Figure 7 and Figure A2. Fine tuning these models using the same data samples after federated training is able to help bridge the gap between the base model performance at a low communication rate and the base model performance at a high communication rate. This is especially evident in the case of clients who are never seen during training. However, for several brain sites, the model overfits during fine tuning, as shown in Figure A2 in Appendix A. This likely happens because brain sites are the majority sample distribution at training time, leading to reconstruction performance already being saturated before any fine tuning.

Figure 8 displays the output of the network $D(\cdot ;\Theta_{k})$ after every two unrolls in the reconstruction, for an unseen site, immediately after federated training (Scaffold), as well as after fine-tuning (Scaffold + personalization). Artifacts are amplified in the baseline model, and personalization helps reduce their intensity.

## 5. Discussion

To train powerful data-driven reconstructions, it is crucial to procure large training datasets which faithfully represent the test data distribution. Due to privacy and regulatory constraints concerning sharing medical data, it may be infeasible for clinics to accrue a dataset locally which is both large enough and has the same distribution as the desired test data. Traditional federated learning attempts to solve the former in a privacy-preserving fashion by only sharing model weights across client institutions via model averaging. As data are often heterogeneous in an inter-client sense, client drift can inhibit the resulting federated model. A number of methods tailored to MRI reconstruction have been recently proposed to tackle this challenge, primarily by constraining the latent space of a feature representation either explicitly [9] or implicitly [25].

Unrolled methods are a powerful class of data-driven approaches for solving medical imaging inverse problems, and in a centralized scenario, they provide good reconstruction quality with as few as 250 training slices (Figure 2). This characteristic motivates their use in federated scenarios where local data are limited in a meaningful sense. It is not immediately clear how to apply latent-space constrained methods to unrolled algorithms. Alternative reconstruction methods based on deep generative priors have recently shown competitive performance with end-to-end unrolled models [43,49,50], and have a proclivity to be more robust to distribution shifts. FedGIMP [23] in particular is a promising federated approach based on deep generative priors but requires adversarial training, which carries with it challenges, such as large training set sizes. Reconstruction times for generative models may also be an issue.

Recent work investigated the low-data regime for single-pass, supervised U-Net reconstruction of the RSS image directly, where it was shown that this approach requires at least thousands of local samples to saturate reconstruction performance [41]. We validate this trend in Figure A1, though we do not reach the saturation point in our experiments. This is in contrast to our findings on unrolled end-to-end models, where performance saturates when using as few as 100 local training samples. FL-MRCM shows a large decrease in performance compared to federated end-to-end approaches, which can likely be attributed to the single-pass U-Net reconstruction backbone it relies on and the same observation for the low-data regime.

At its core, federated learning seeks to provide a model that generalizes well to all clients (participating and not participating). This was the driving motivation for adaptive methods [18,19]. Table 2 shows that adaptive approaches at training time have little impact in the scenario where all clients are homogeneous (i.i.d.). Adaptivity becomes more important in the heterogeneous (non-i.i.d.) case as we show in Table 3 and Figure 3. Table 3 shows the average validation performance evaluated on data from the same type that the client had access to during training. Adaptive approaches surpass the more simplistic weight aggregation in FedAvg, due to their use of auxiliary optimization variables at training time, designed to combat client drift. Not only do adaptive approaches surpass FedAvg in the non-i.i.d. scenario, they also often outperform the centralized model for high communication rates (Table 3). This finding can be explained with the observation that, in the process of averaging individual models, each model is implicitly undergoing a form of regularization. Unlike the federated setting, the centralized setting we utilize does not use make use of any assumption of data heterogeneity, which may sometimes place its performance below that of adaptive federated learning algorithms that are designed with data heterogeneity in mind. This could in turn provide an approach to improve the performance of the centralized approach via a multi-task formulation, such as the one investigated in [28].

Another aspect in federated learning is the effect of a limited communication budget. When training is performed synchronously, global updates occur when at least a subset of individual clients are ready. This can create a communication bottleneck that restricts the number of global update rounds that occur during the training process. We found that limited communication between clients and the global server has an adverse effect for clients who are actively participating in the federated training. Interestingly, we find that a new client’s performance is almost agnostic to the communication rate, with the important caveat that reconstructions for novel clients at various communication rates may not be of diagnostic quality (Figure 5).

To overcome the performance drop experienced by participating clients in communication constrained settings, as well as to novel clients, we investigated simple personalization through model fine tuning. We observed that fine tuning enabled performance boosts both for under-represented clients participating in training and for those that did not participate in federated training at any time (Figure 7). We found that 50 slices with five-fold cross-validation were sufficient to fine tune the model at each site.

In this work, we made several design choices, which do not cover all federated learning scenarios. The simplifications we assumed were synchronous training between clients, and no client drop-out at each communication round. We also assumed a single fixed acceleration factor and sampling pattern for all data. In future work, it will be crucial to investigate both the effects of asynchronous learning, varying levels of client dropout, and differences in scan protocol. Additionally, a promising future research direction is to investigate the performance of larger client pools (≫10), varying dataset sizes at each site, and varying scanning conditions, such as acceleration factor.

## 6. Conclusions

Due to their need for relatively small training set sizes when compared to feed-forward networks or generative model based reconstruction methods, unrolled networks are an interesting application of federated MRI reconstruction. With this in mind, we explored i.i.d. and non-i.i.d. federated learning scenarios, with realistic data distributional shifts (anatomy, contrast, field strength) and varying communication budgets. Additionally, we showed that by personalizing federated unrolled networks through fine tuning on limited data, we were able to boost the performance of both under-represented clients present in federated training and clients absent at federated training time. Our experimental results show that federated learning using unrolled networks can help bridge the gap between low resource areas and large institutions by enabling smaller institutions to utilize state-of-the-art deep learning methods without a large number of local samples for model training and tuning.

## Figures and Tables

**Figure 1 bioengineering-10-00364-f001:**
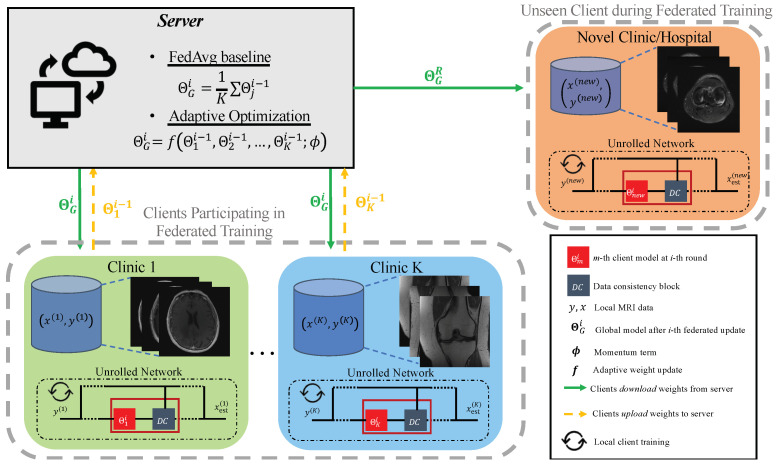
High-level block diagram of federated unrolled optimization for accelerated MRI reconstruction. Local data are not shared with the server or between participating clients. After federated training, all clients (seen and unseen at federated training time) receive weights $\Theta_{G}^{R}$ and perform fine tuning using only local samples.

**Figure 2 bioengineering-10-00364-f002:**
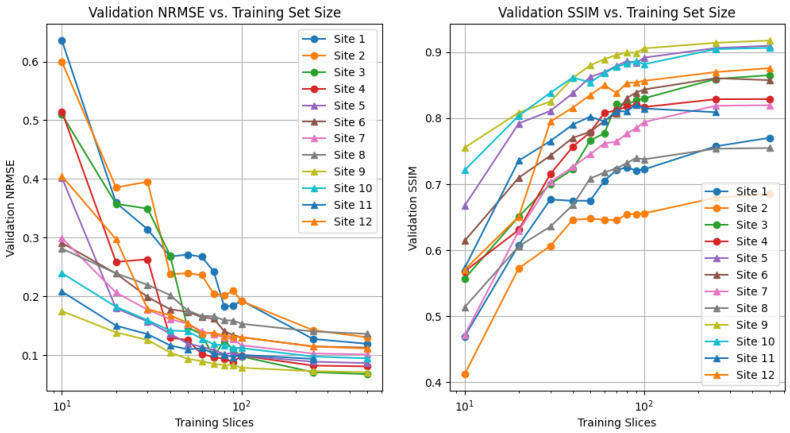
Validation normalized root mean squared error (NRMSE, (**left**)), and structural similarity index measure (SSIM, (**right**)) for varying number of local training samples (2D slices) in centralized training scenario. All fastMRI sites were trained and tested individually.

**Figure 3 bioengineering-10-00364-f003:**
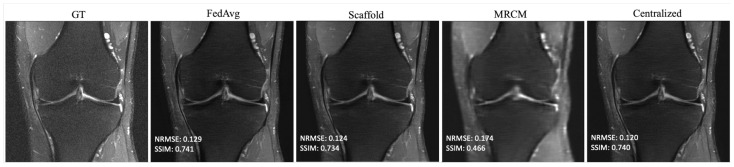
Example reconstructions for knee PDFS 1.5T obtained in the i.i.d. client (knee PDFS 1.5T) scenario, and 240 communication rounds.

**Figure 4 bioengineering-10-00364-f004:**
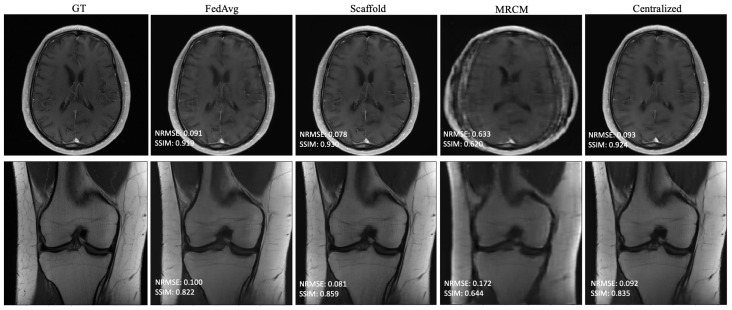
Example reconstructions obtained in the non-i.i.d. client scenario with 240 communication rounds: (**Top**) Brain T1-POSTCON 3T, (**Bottom**) Knee PD 3T.

**Figure 5 bioengineering-10-00364-f005:**
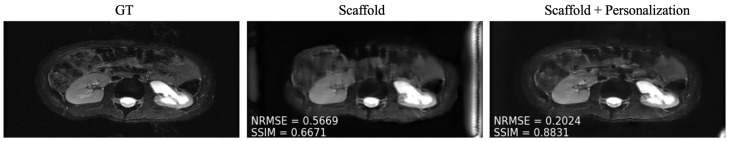
Example reconstructions on a new client (Stanford) using (**Center**) only the pre-trained Scaffold model, (**Right**) Scaffold + personalization.

**Figure 6 bioengineering-10-00364-f006:**
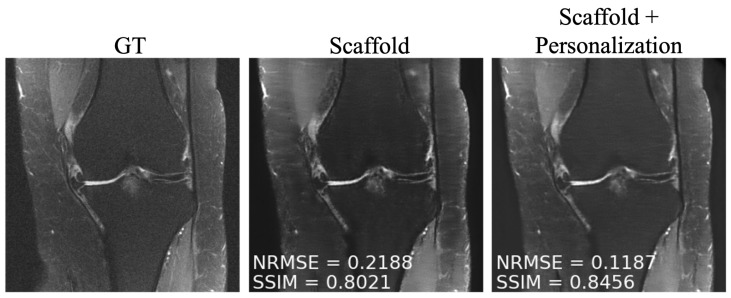
Example reconstructions on a participating client (site 2), with four communication rounds, using (**Center**) only the pre-trained Scaffold model, (**Right**) Scaffold + personalization.

**Figure 7 bioengineering-10-00364-f007:**
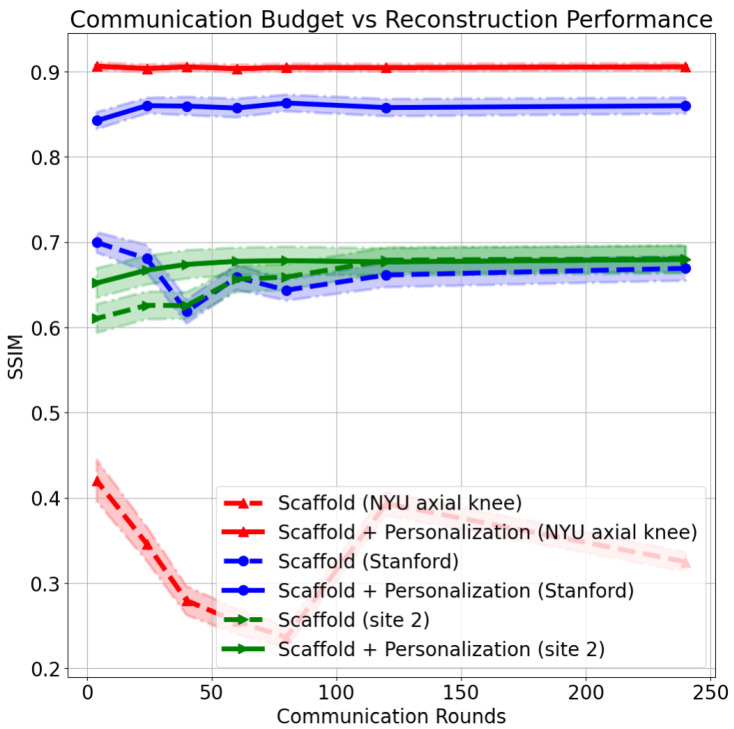
Structural similarity index measure (SSIM) comparison between baseline (Scaffold only) and Scaffold + personalization, as communication budget varies, for three different distributions: NYU axial knee (unseen at federated training), Stanford (unseen at federated training), and PDFS knee (seen in federated training).

**Figure 8 bioengineering-10-00364-f008:**
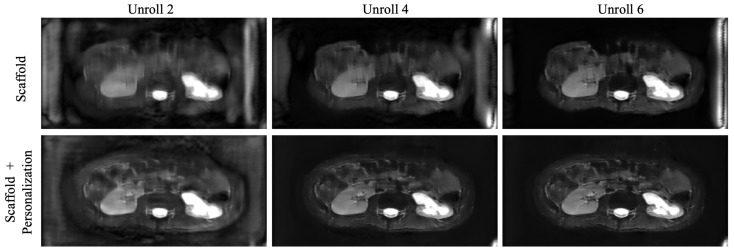
Images across unrolls (output of $D(\cdot ;\Theta_{k})$) for: (**Top**) Scaffold initialized model, and (**Bottom**) Scaffold + personalization model applied to an unseen client distribution.

**Table 1 bioengineering-10-00364-t001:** fastMRI client list.

Site	Anatomy	Contrast	Field Strength	IID Training	Non-IID Training
1	Knee	PDFS	3T	Unseen	Present(×1)
2	Knee	PDFS	1.5T	Present(×10)	Present(×1)
3	Knee	PD	3T	Unseen	Unseen
4	Knee	PD	1.5T	Unseen	Present(×1)
5	Brain	T2	3T	Unseen	Present(×1)
6	Brain	T2	1.5T	Unseen	Present(×1)
7	Brain	FLAIR	3T	Unseen	Present(×1)
8	Brain	FLAIR	1.5T	Unseen	Present(×1)
9	Brain	T1 POSTCON	3T	Unseen	Present(×1)
10	Brain	T1 POSTCON	1.5T	Unseen	Present(×1)
11	Brain	T1 PRECON	3T	Unseen	Present(×1)
12	Brain	T1 PRECON	1.5T	Unseen	Unseen

**Table 2 bioengineering-10-00364-t002:** i.i.d. performance across 10 sites. The average loss values over entire validation dataset are shown for the structural similarity index measure (SSIM) and normalized root mean squared error (NRMSE). Top performing values are shown in bold font.

Algorithm	SSIM ↑	Increase	NRMSE ↓	Decrease
Centralized	0.708	–	0.135	–
FedAvg	0.710	+0.28%	0.134	−1%
FedAdam	**0.713**	+0.71%	0.151	+12%
FedYogi	0.712	+0.57%	0.138	+2%
FedAdaGrad	0.712	+0.57%	0.139	+3%
Scaffold	0.711	+0.42%	**0.130**	−4%
FL-MRCM	0.489	−31%	0.167	+23%

**Table 3 bioengineering-10-00364-t003:** Non-i.i.d. performance for 12 fastMRI clients. The average loss values over entire validation dataset are shown for the structural similarity index measure (SSIM) and normalized root mean squared error (NRMSE). The best performance (highest SSIM, lowest NRMSE) is shown in bold font. For all sites, adaptive algorithms have the best performance.

**Algorithm**	**Site 1**		**Site 2**		**Site 3**		**Site 4**		**Site 5**		**Site 6**	
	SSIM	NRMSE	SSIM	NRMSE	SSIM	NRMSE	SSIM	NRMSE	SSIM	NRMSE	SSIM	NRMSE
FedAvg	0.767	0.142	0.653	0.229	0.824	0.111	0.844	0.089	0.919	0.107	0.900	0.109
FedAdam	0.768	0.142	0.677	0.183	0.821	0.117	0.845	0.093	0.928	**0.096**	0.901	**0.104**
FedYogi	0.770	0.140	0.675	0.192	0.826	0.105	0.848	0.090	0.922	0.102	0.900	0.106
FedAdaGrad	0.766	0.144	0.675	0.190	0.826	0.113	0.846	0.092	0.919	0.109	0.898	0.109
Scaffold	**0.771**	**0.134**	**0.680**	**0.166**	**0.838**	**0.098**	**0.848**	**0.083**	**0.929**	0.098	**0.902**	0.107
FL-MRCM	0.609	0.208	0.458	0.230	0.670	0.193	0.682	0.185	0.806	0.266	0.761	0.294
Centralized	0.762	0.142	0.674	0.172	0.824	0.118	0.836	0.095	0.917	0.116	0.892	0.127
**Algorithm**	**Site 7**		**Site 8**		**Site 9**		**Site 10**		**Site 11**		**Site 12**	
	SSIM	NRMSE	SSIM	NRMSE	SSIM	NRMSE	SSIM	NRMSE	SSIM	NRMSE	SSIM	NRMSE
FedAvg	0.872	0.098	0.803	0.146	0.920	0.086	0.925	0.106	0.851	0.106	0.905	0.113
FedAdam	**0.874**	**0.097**	0.814	0.133	0.926	**0.079**	0.928	0.100	0.853	0.105	0.904	**0.108**
FedYogi	**0.874**	**0.097**	**0.815**	**0.129**	0.926	**0.079**	**0.929**	**0.098**	0.856	0.100	**0.906**	**0.108**
FedAdaGrad	0.873	**0.097**	0.811	0.137	0.924	0.082	**0.929**	0.105	0.854	0.103	0.903	0.111
Scaffold	0.872	**0.097**	0.812	0.131	**0.927**	0.082	**0.929**	0.102	**0.859**	**0.098**	0.903	0.112
FL-MRCM	0.568	0.278	0.521	0.258	0.757	0.370	0.749	0.293	0.743	0.211	0.738	0.284
Centralized	0.863	0.107	0.800	0.145	0.920	0.097	0.920	0.121	0.852	0.107	0.897	0.134

**Table 4 bioengineering-10-00364-t004:** Fine-tuning hyper parameter result.

Client	Non-i.i.d. Training	$r_{\mathrm{fine}}$	$N_{\mathrm{fine}}$
Site 2	Present	$1\times 10^{-5}$	170
Stanford	Unseen	$1\times 10^{-4}$	200
NYU axial knee	Unseen	$1\times 10^{-4}$	200

## Data Availability

In all our experiments, we used publicly available datasets. These are available through the fastMRI project at https://fastmri.org/ (accessed on 1 March 2022), as well as the open repository at http://mridata.org/ (accessed on 1 March 2022). Code to reproduce results in this work will be made available at https://github.com/utcsilab/Unrolled_FedLrn (accessed on 6 February 2023) following publication.

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
