# Peer review of "Federated End-to-End Unrolled Models for Magnetic Resonance Image Reconstruction"

_bioengineering, 2023, doi:10.3390/bioengineering10030364_

Round 1

Reviewer 1 Report

Complete, well-presented and well-organized article

I suggest to replace the paragraph title "Related Work" with "Background"

I suggest a careful revisions of the captions. Both in figure and in table captions, all the abbreviations should be clearly re-explained (NRMSE and SSIM)

Author Response

I suggest to replace the paragraph title "Related Work" with "Background"

We thank the reviewer for their suggestion and agree that this is a more suitable title. We have edited the title of Section 1.1. accordingly.

I suggest a careful revisions of the captions. Both in figure and in table captions, all the abbreviations should be clearly re-explained (NRMSE and SSIM)

We thank the reviewer for their suggestion, we have now included explanations of the used abbreviations in the captions of the relevant figures and tables (Figures 2, 3, 7 and Table 3).

Reviewer 2 Report

In this paper, the authors investigated the application of federated learning to MRI image reconstruction.

Experiments showed that the adaptive optimization algorithm produces well reconstructed images even with limited data.

This is a very interesting work, and I consider it to be acceptable if the following minor comments are improved.

1. please adjust the image quality of Figure 3. The knee structure is difficult to see.

2. Are SSIM and NRMSE sufficient as the only image quality evaluation indices? Please also consider PSNR, etc.

Author Response

  1. please adjust the image quality of Figure 3. The knee structure is difficult to see.

We thank the reviewer for their suggestion and have adjusted the brightness/contrast for Figure 3 to help make finer structures within the knee easier to see. 

  1. Are SSIM and NRMSE sufficient as the only image quality evaluation indices? Please also consider PSNR, etc.

We have included an additional table in the appendix (Table 1A) containing PSNR values for our Non-IID experiments (originally shown in Table 3). We have not included FL-MRCM in this table due to its extremely poor performance.

Reviewer 3 Report

This paper evaluates various federated learning algorithms, for learning end-to-end unrolled MRI reconstruction in a low data regime. In particular, 12 sites are simulated, with each site containing data of a particular anatomy/contrast/field strength combination, such that sites have heterogeneous data with respect to each other.

The analysis is generally comprehensive. However, some issues might be addressed:

1. For the non-iid results as reported in Table 3, it might be clarified as to what the test data for each site was, for the results reported in Table 3. In particular, does each site test on only the type of scans that it had access to (i.e. site, anatomy, contrast, field strength, as in Table 1), or on all combinations of scan types?

2. Moreover, while 50 local scans were available for training at each site as determined in Section 4.1 for the low data regime, it is not clear how many scans were available for testing. Five-fold cross-validation on training data is described in Section 3.4, but seemingly only for hyperparameter selection. This might be clarified.

3. For the Centralized algorithm, it might be more explicitly stated as to what data each site has access to - is it all training data from all sites? In particular, it is unclear why adaptive algorithms would outperform such a centralized algorithm (which has all data).

4. Table 3 appears to report the non-i.i.d results without fine-tuning, with fine-tuning analysis limited to only a few clients (Table 4). This might be justified, since it appears that fine-tuning for a site's particular type of scans would have the potential to significantly improve performance (on that type of scan).

5. In Line 327, the Figure number might be included.

Author Response

  1. For the non-iid results as reported in Table 3, it might be clarified as to what the test data for each site was, for the results reported in Table 3. In particular, does each site test on only the type of scans that it had access to (i.e. site, anatomy, contrast, field strength, as in Table 1), or on all combinations of scan types?

We thank the reviewer for pointing this out, Table 3 shows test performance only on the type of scans that the site had access to, which is deemed most relevant for them. We have clarified this aspect in Section 5.

  1. Moreover, while 50 local scans were available for training at each site as determined in Section 4.1 for the low data regime, it is not clear how many scans were available for testing. Five-fold cross-validation on training data is described in Section 3.4, but seemingly only for hyperparameter selection. This might be clarified.

We thank the reviewer for their comments. At a local client, after receiving the federated network weights, the goal is to fine-tune the model on the local data distribution using all available samples at that particular client. However, to do this we need to ensure that we can find both the early stopping point and correct learning rate with only the limited data available to the client. So for each client we partition the 50 samples into two non-overlapping subsets: 40 training slices, and 10 validation slices. We then fine-tune the model on the 40 training slices and validate the performance on the remaining 10 holdout slices. We repeat this process for the same client by partitioning the 50 slices in this way 4 more times, each time retraining the model with 40 slices and validating on 10 slices. After completing this process five times for various learning rates and epochs we select the best stopping point and learning rate based on the validation results from each model tested (five in total). Next we take all 50 slices available at the same client and fine-tune the model using the learning rate and stopping point selected previously. The test set is the same as was used for the other results presented. 

  1. For the Centralized algorithm, it might be more explicitly stated as to what data each site has access to - is it all training data from all sites? In particular, it is unclear why adaptive algorithms would outperform such a centralized algorithm (which has all data).

We thank the reviewer for their comments. Indeed, in the centralized setting, we assume there is effectively a single site which has access to the training data from all the sites that arise in federated learning. We have clarified this in Section 2.2.

As to why it is possible for adaptive federated learning algorithms to slightly surpass the centralized algorithm in non-iid scenarios (such as in Table 3), this can be explained by the fact that unlike the federated setting, the centralized setting we utilize does not use make use of any assumption of data heterogeneity, which may sometimes place its performance below that of adaptive federated learning algorithms that are designed with data heterogeneity in mind (such as Scaffold).

This implies that we could have obtained a better centralized method by taking data heterogeneity in account - for example, using a multi-task learning setting such as the one in [1] - but this would have been out-of-scope for the federated setting. We have included this discussion and reference in Section 5.

[1] Liu, V.; Ryu, K.; Alkan, C.; Pauly, J.M.; Vasanawala, S. Multi-Task Accelerated MR Reconstruction Schemes for Jointly Training Multiple Contrasts. In Proceedings of the NeurIPS 2021 Workshop on Deep Learning and Inverse Problems, 2021.

  1. Table 3 appears to report the non-i.i.d results without fine-tuning, with fine-tuning analysis limited to only a few clients (Table 4). This might be justified, since it appears that fine-tuning for a site's particular type of scans would have the potential to significantly improve performance (on that type of scan).

We thank the reviewer for their comment, and would like to point to Figure A2 in the Appendix, which already contains results for fine-tuning all sites (except ones already present in Table 4), as a further function of communication budget.

Figure A2 shows the average validation SSIM for each site (evaluated on that site’s validation data only), as a function of the number of federated communication rounds, both for the regular case and the fine-tuned (personalized) case. Indeed, as the reviewer pointed out, it can be seen that for 6 out 10 sites there is a fine-tuning performance gain regardless of communication rate, though in some cases fine-tuning can reduce performance (even on that type of scan). The discussion in the Appendix captures the above aspects.

  1. In Line 327, the Figure number might be included.

We thank the reviewer for noticing this, we have corrected the unknown symbol and the line now correctly refers to “Figure 7 and Figure A2”.

Round 2

Reviewer 3 Report

We thank the authors for addressing our previous comments.